# Dispersoids in Al-Mg-Si Alloy AA 6086 Modified by Sc and Y

**DOI:** 10.3390/ma16082949

**Published:** 2023-04-07

**Authors:** Franc Zupanič, Sandi Žist, Mihaela Albu, Ilse Letofsky-Papst, Jaka Burja, Maja Vončina, Tonica Bončina

**Affiliations:** 1Faculty of Mechanical Engineering, University of Maribor, Smetanova 17, SI-2000 Maribor, Slovenia; 2Impol 2000 d.d., Partizanska ulica 38, SI-2310 Slovenska Bistrica, Slovenia; 3Graz Centre for Electron Microscopy, Steyrergasse 17, A-8010 Graz, Austria; 4Institute of Electron Microscopy and Nanoanalysis and Centre for Electron Microscopy, Graz University of Technology, NAWI Graz, Steyrergasse 17, A-8010 Graz, Austria; 5Institute of Metals & Technology, Lepi pot 11, SI-1000 Ljubljana, Slovenia; 6Faculty of Natural Sciences and Engineering, University of Ljubljana, Aškerčeva cesta 12, SI-1000 Ljubljana, Slovenia

**Keywords:** aluminium, alloy, heat treatment, microstructure, dispersoids, dilatometry, DSC, hardness

## Abstract

The aluminium alloy AA 6086 attains the highest room temperature strength among Al-Mg-Si alloys. This work studies the effect of Sc and Y on the formation of dispersoids in this alloy, especially L1_2_-type ones, which can increase its high-temperature strength. A comprehensive investigation was carried out using light microscopy (LM), scanning (SEM), and transmission (TEM) electron microscopy, energy dispersive spectroscopy (EDS), X-ray diffraction (XRD), differential scanning calorimetry (DSC), and dilatometry to obtain the information regarding the mechanisms and kinetics of dispersoid formation, particularly during isothermal treatments. Sc and Y caused the formation of L1_2_ dispersoids during heating to homogenization temperature and homogenization of the alloys, and during isothermal heat treatments of the as-cast alloys (T5 temper). The highest hardness of Sc and (Sc + Y) modified alloys was attained by heat-treating alloys in the as-cast state in the temperature range between 350 °C and 450 °C (via T5 temper).

## 1. Introduction

The aluminium alloys 6xxx are based on the Al-Mg-Si system [1]. They generally possess moderate strength, excellent formability, and high ductility. They are widely used in automotive, aerospace, and other industries [2,3]. The strengthening phases are GP-zones and Mg_2_Si-type metastable precipitates, such as β′ and β″ [4]. The Cu addition can promote the formation of the strengthening Q′-AlCuMgSi precipitates [5,6,7]. The precipitation of Q′-AlCuMgSi precipitates occurs via precursor phases, making precipitation rather a complex process [5]. Mao et al. [6] determined the growth kinetics and shape of β″ precipitates by combining phase-field modelling and experimental studies. In the Al-Mg-Si alloys, the addition of Zr caused grain size reduction during solidification and promoted the formation of an equiaxed grain structure [8]. Zirconium additional causes strengthening and resistance to grain coarsening by forming tetragonal and L1_2_ Al_3_Zr dispersoids [9,10]. Jia et al. [11] found a negative effect of Zr addition in Al-Mn-Zr alloys because Zr caused coarsening of Mn-rich dispersoids.

Aluminium alloys with minor Sc and Y additions are utilized in the military, aircraft, and marine industries because of their corrosion resistance and high strength. They can also be found as construction elements for high-performance sports equipment, such as bicycle frames [12]. The characteristics of Sc-modified Al-alloys are described in detail in several review articles [13,14,15,16]. Scandium causes substantial grain refinement in almost all Al-alloys [17]. The primary Al_3_Sc has an L1_2_ crystal structure with a very small lattice misfit with the α-Al solid solution, acting as a potent heterogeneous nucleation site for α-Al during solidification [18]. Sc in the Al-Mg-Si alloy also significantly decreases the secondary dendrite arm spacing (SDAS) [19]. Zirconium additions cause layered primary Al_3_(Sc,Zr) phase formation during solidification [18,20]. In the solid state, L1_2_ precipitates cause dispersion strengthening and improve the high-temperature stability of diluted Al-Sc, Al-Zr, and Al-Sc-Zr alloys [21]. Precipitation of L1_2_-Al_3_Sc starts between 200 °C and 250 °C, and L1_2_-Al_3_Zr between 350 °C and 375 °C [20]. The core–shell L1_2_ precipitates (core is Sc rich, whose shell is Zr rich) strongly enhance the thermal stability of alloys [20,22,23,24]. The small amount of Si strongly accelerates the nucleation of L1_2_-Al_3_(Sc,Zr) precipitates, and silicon replaces Al in precipitates [25]. The formation of L1_2_-precipitates is affected by microsegregation, and Fe and Si clusters facilitate the nucleation of L1_2_-precipitates [26]. Adding a higher amount of Si is detrimental because in the alloys containing more than 0.078% Sc and 0.18% Si V-phase AlSc_2_Si_2_ can form. It adversely affects the strengthening because it decreases the contents of Sc and Si in the α-Al solid solution, resulting in a smaller volume fraction of desirable L1_2_-Al_3_X dispersoids [27]. A combination of small Zr and Sc alloying additions to AA 6106 combinations resulted in a fine grain structure close to the average grain size of 5 μm in AA 6106 with 0.5% Sc [28]. These Al_3_Zr and Al_3_(Sc,Zr) dispersoids stabilise the microstructure at higher temperatures and inhibit grain growth [29]. They impede the movement of grain boundaries, and lead to a fine grain structure after recrystallisation or even preserve the non-recrystallised structure after homogenisation [15].

The yttrium-modified Al-alloys were less investigated than Sc- and Zr-modified alloys. Yttrium strongly reduces grain sizes in Al-Si alloys during solidification [30]; thus, an Al-Y-B grain refiner was developed [31]. It also modifies the eutectic silicon and iron-rich phases in Al-Si-Cu and Al-Si piston alloys [32,33,34]. Shen et al. [35] investigated the effect of Y on the properties of Al-Mg-Mn alloys. The strength and ductility increased up to 0.3% Y, then started deteriorating due to the formation of coarser Al_2_Y particles. Guo et al. revealed an essential influence of Zr, Y, and Y + Zr additions to Al-Mn-Cu alloys [36]. The combined additions of Zr and Y in the range 0.1–0.3 wt. % produced the best properties. Wang et al. [37] studied the effect of some transition metals, including Y, on an Al-Mg-Si alloy. The additions of Y to AA 5083 reduced the grain size and improved mechanical properties [38]. They found that the most Si was consumed for the formation of the dispersoid in a Y-added alloy, which reduced the amount of Si in the solid solution for the formation of strengthening precipitates. An increase in mechanical properties was also achieved by the combined Y, Sc, and Zr addition to an Al-Mg alloy [39]. Pozdiakov et al. [40] found L1_2_-Al_3_(Sc,Y) precipitates in Al-Sc-Y alloys heat-treated at 400 °C.

There are several dispersoid phases in commercial Al-Mg-Si alloys. One of the essential dispersoid phases is α-AlMnSi because Mn is present virtually in all Al-Mg-Si alloys leading to the formation α-AlMnSi. This phase can also dissolve other elements, such as Cr and Fe [41,42]. This type of dispersoid can effectively pin migration of dislocation and grain boundaries. Thus, they restrain recovery and recrystallization [43] and increase the flow stress of Al-alloys at elevated temperatures [44]. Their properties can be improved by Zr-additions [45,46]. Remoe et al. [47] revealed that in Al-Mg-Si alloys with different Mn-contents, the heating rates affect the density and spatial distribution of α-AlMnSi dispersoids. It was found that the additions of Cd [48] and Mo [49] can enhance the elevated properties of 3xxx alloys. Thus, it would be viable to verify the effects of different elements on the distribution of α-AlMnSi in 6xxx alloys. The formation of α-AlMnSi induced by microadditions of Mo and Mn to dilute Al-Sc-Zr-Re-Si can enhance strengthening caused by L1_2_-precipitates [50].

The alloy AA 6086 is a wrought alloy used for extruded products in demanding automotive industry applications, such as forged steering rods, developed recently. It was developed from AA 6082 by modifying its chemical composition, mainly by changing the ratio Si:Mg and adding some Zr and Cu [51]. Its tensile strength can exceed 480 MPa in T6 temper, while the ductility still surpasses 10%. It also exhibits excellent fatigue properties [52].

Our previous work [53] evaluated the impact of Sc additions on the microstructural evolution during solidification and by precipitation hardening (T6 temper) of the alloy AA 6086. Only a little attention was given to dispersoids in the microstructure. It is known that a combination α-AlMnSi and L1_2_-precipitates can contribute to higher heat resistance [53]. The resistance to grain growth during recrystallization and high-temperature strength strongly depends on the formation and distribution of dispersoids. Thus, the main aim of this work is to study the effects of minor Sc and Y additions, and the heat treatment on the formation and distribution of dispersoids formation. In addition to microstructural analysis, a significant focus was given to differential scanning calorimetry and dilatometry to discern the mechanisms and kinetics of dispersoid formation.

## 2. Materials and Methods

### 2.1. Synthesis of Alloys and Chemical Compositions

Table 1 gives the chemical compositions of the investigated alloys. The reference alloy has a typical composition for AA 6086 alloy. The Si:Mg ratio is around 1.6; it contains about 0.7% Mn to provide a sufficient fraction of α-AlMnSi dispersoids and 0.17% Zr to form Al_3_Zr dispersoids. The addition of Sc is around 0.2%, which is very often in Al-alloys since it provides L1_2_-Al_3_Sc precipitates, and in combination with Zr, core–shell Al_3_(Sc,Zr) precipitates, which have better heat resistance [21]. Only a tiny addition of Y was selected. With this small addition, Y is not expected to form its phases, but will enter into the Al_3_Zr and Al_3_Sc phases. The total amounts of Sc and Y are low to reduce the fraction of detrimental AlSc_2_Si_2_ phase, which may form during heat treatment.

The alloys were melted using commercially available Al99.8, AlMn10, AlCu25, AlSc2, AlY10, and AlZr10. The alloys were cast into copper moulds with cylindrical segments with diameters of 10 mm.

The alloys were heat treated in an electric-resistant furnace Bosio (Bosio, Bukovžlak, Slovenia) to study the mechanism and kinetics of dispersoids formation. The formation of dispersoids was studied in two starting conditions of samples: namely, in the as-cast and homogenized conditions. They were exposed to different temperatures for up to 5 h to study the formation of dispersoids. The as-cast samples were put into the furnace, already heated to the selected heat-treating temperature (200 °C, 250 °C, 300 °C, 350 °C, 400 °C, or 450 °C), and held at this temperature for up to 5 h. Finally, they were quenched in water to room temperature. The homogenized samples were held at 520 °C for 6 h, quenched in the water, then isothermally heat treated at 350 °C, 400 °C, and 450 °C for up to 5 h. The heat-treating procedures were similar to that in industrial practice.

### 2.2. Characterization of Alloys

The alloys were sectioned using two metallographic saws Labotom 5 (Struers, Ballerup, Denmark) and IsoMet 1000 (Buehler, Lake Bluff, IL, USA). The light (LM) and scanning electron (SEM) microscopy samples were mechanically ground and polished. After final polishing using a 3 μm diamond paste, the microstructure was revealed by chemical etching with Weck’s reagent consisting of 2 g KMnO_4_ (Merck KGaA, Darmstadt, Germany), 1 g NaOH (Merck KGaA, Darmstadt, Germany), and 50 mL of distilled water.

The fundamental metallographic analysis was performed by light microscope Neophot 300 (Nikon, Tokyo, Japan) and scanning electron microscopy using scanning electron microscopy JSM-IT800 SHL (JEOL Ltd., Tokyo, Japan) and Sirion 400 NC (FEI, Eindhoven, The Netherlands) equipped with an energy-dispersive spectrometer (Oxford Analytical, Bicester, UK).

XRD (X-ray Diffraction) using synchrotron X-rays was used for the phase analysis. A detailed description can be found in Ref. [54].

The differential scanning calorimetry (DSC) was performed using Netzsch, STA 449c Jupiter (The Netzsch Group, Selb, Germany). The samples were in the as-cast condition; the typical size was 4 mm in diameter and height 2 mm. The dilatometry tests were carried out in a quenching/deformation dilatometer DIL 805A/D (TA Instruments, New Castle, DE, USA) using cylinder specimens ϕ = 4 mm, *L* = 10 mm). The temperature regime was the same in both tests and is shown in Figure 1. The first regime was carried out to follow processes during isothermal annealing (Figure 1a) of the alloys in the as-cast condition. On the other hand, the second regime was applied to study processes during heating to homogenization temperature 520 °C, during homogenization (520 °C), and the processes that take place in the homogenized condition at 400 °C, at the temperature at which L1_2_-Al_3_X precipitates are expected to form (Figure 1b). It was not possible to attain high cooling rates in the DSC-apparatus. Thus, it was impossible to simulate quenching from the homogenization temperature to room temperature, which can retain higher fractions of alloying elements in the aluminium solid solution. Nevertheless, the conditions by DSC and dilatometry were the same, thus allowing a direct comparison of the results. The specimen tested by dilatometry were compression tested at the same device at room temperature up to 0.7 true strain.

Vickers hardness measurements HV 30 (load 294.2 N, loading time 20 s, holding at maximum load 10 s, unloading time 1 s) were used for determining the effects of heat treatment (Duramin-40 M2, Struers, Ballerup, Denmark). This load produced sufficiently large indentations to test several crystal grains, and the results did not scatter due to local differences in the microstructure.

Lamellas for transmission electron microscopy (TEM) were prepared by electrolytic thinning. High-resolution TEM Titan^3^ G2 60–300 (FEI, Eindhoven, The Netherlands) and energy-dispersive X-ray spectroscopy (SuperX, Bruker, Billerica, MA, USA) were used for investigating the samples alloys with Sc and Y in the homogenized condition (520 °C, 6 h, water quench), and homogenized and isothermally annealed for 6 h at 400 °C. The images presented in this work were mainly taken in STEM mode with a high-angle annular dark field detector (HAADF detector), in which phases with higher atomic number elements appear brighter.

## 3. Results

### 3.1. Initial as-Cast and Homogenized Conditions

Figure 2 shows the microstructures of all alloys in the as-cast state and after homogenization. The detailed microstructural analysis of the reference alloys 6086 and 6086-Sc can be found in previous publications [51,55]. The microstructures of the alloys in the as-cast state were similar at this lower magnification due to small additions of Sc and Y. The dendritic solid solution α-Al grains prevailed in the microstructure, and other phases were present in the interdendritic regions. The brightest Θ-Al_2_Cu phase was present in islands which solidified at the end of solidification. The brighter α-AlMnSi phase also contained some Cr and Fe, while the darker Mg_2_Si phase contained only Mg and Si. The tetragonal Al_3_X (t-Al_3_X) particles were rather small. They contain only Al and Zr in the reference alloy, but in 6086-Sc and 6086-ScY alloys, they also contained Sc and Y. In addition to these phases, small fractions of minor phases, such as Q-AlCuMgSi and ZrSi_2_, were also identified. No particles were observed inside the aluminium-rich solid solution. The α-Al matrix was supersaturated with the alloying elements and vacancies due to rather high cooling rates during solidification, which were higher than reported in the previous paper [55].

During homogenization α-AlMnSi hardly changed, Mg_2_Si partially dissolved, and the undissolved remains became larger and spheroidal. Islands of Θ-Al_2_Cu disappeared, and minor phases wholly dissolved in the matrix. Inside the α-Al, plenty of different dispersoids formed that will be presented later in greater detail.

The comparison of the XRD patterns is shown in Figure 3. XRD revealed only phases with higher volume fractions (α-Al, α-AlMnSi, Mg_2_Si, and Θ-Al_2_Cu). The peaks mainly remained at the same position, while the peaks of Θ-Al_2_Cu disappeared, indicating its dissolution in the Al-matrix, which was also observed by microstructural analysis.

Figure 4 and Figure 5 show the results of DSC and dilatometry, which may give some information regarding the nature of processes taking place during the heating of the alloys from RT to 520 °C and holding at the homogenization temperature.

The processes, during heating to 520 °C, were endothermic, indicating the heat flow to the specimen (negative values). Endothermic reactions’ intensity rapidly fell to 50 °C, then the falling rate started to decrease, reaching the minimum value near 80 °C for the reference sample and around 90 °C for the 6086-Sc and 6086-ScY. In this temperature range, clusters and GP zones may form [56]. At further heating, three exothermic peaks occurred (Figure 4a). The first exothermic peak extended from 90 °C to 210 °C (Peak A). This peak is attributed to the formation of metastable precipitates in Al-Mn-Si alloys, namely, GP-zones and β″-precipitates, while in the Cu-modified alloys; additionally, metastable Q′-AlCuMgSi precipitates may form. The second peak (Peak B) lay between 220 °C and 330 °C. This peak is related to forming metastable β′-precipitates and stable β-Mg_2_Si [57] and Q-AlCuMgSi phases. The third peak appeared between 380 °C and 490 °C (Peak C); it is correlated to the formation of dispersoids in the matrix. In 6086, predominantly to the formation of α-AlMnSi and t-Al_3_X, the additions of Sc and Y did not noticeably affect the shape and size of the first peak and the beginning of the second peak, but altered the behaviour of alloys during further heating. The heat flow dropped only slightly after the second peak, and the start of the third peak was shifted to about 40 °C higher temperatures. The third peak in the reference 6086 alloy was clearly defined, but in the 6086-Sc and 6086-ScY, only a small hump formed, the exothermic reaction still proceeded after it up to the 520 °C. Clearly defined peak for the reference alloy indicates that almost all dispersoids have formed before attaining the homogenization temperature, while in the Sc and Y modified alloys processes took place continuously.

After temperature stabilization at 520 °C, the reference alloy rapidly fell into the endothermic region (Figure 4b). This observation can imply that no exothermic reactions occurred (formation of dispersoids) and that endothermic reactions prevailed (dissolution of different phases, predominantly Mg_2_Si). On the other hand, by adding Sc and Y, the net thermal effect was always exothermic, indicating almost continuous precipitation of L1_2_-Al_3_X precipitates.

The dilatation curves almost coincided up to 300 °C (Figure 5a). The dilatation of 6086-Sc was about 13% smaller than that of the other two alloys above 300 °C. During holding at 520 °C, the alloy 6086-Sc shrank very fast, while the reference alloy and 6086-ScY shrank similarly (Figure 5b). The shrinkage is caused predominantly by the dissolution of Mg_2_Si, which has a lower density than the Al-matrix. The density of the Al-matrix is approximately 2700 kg/m^3^, while those of the Mg_2_Si is 1990 kg/m^3^ [58]. The much stronger shrinkage of 6086-Sc can be caused by faster precipitation of L1_2_-precipitates having a higher density than the α-Al matrix (Al_3_Sc 3108 kg/m^3^ [59]). The density of α-AlMnSi and t-Al_3_X, which form at the third peak, are 3520–3620 kg/m^3^ [60], and 4170 kg/m^3^ [58], respectively.

Figure 6 shows the TEM micrographs of the alloy 6086–Sc and 6086-ScY after homogenization. The large brighter phase in the microstructure was primary α-AlMnSi predominantly located in the interdendritic regions and at the grain boundaries. The dispersoids were non-uniformly distributed in the matrix. The α-AlMnSi that formed during heating and homogenization had a plate-like shape. The number density of the α-AlMnSi plates was much higher near the interdendritic regions. Only a few very large α-AlMnSi plates were present at the dendrite centres, with lengths of several micrometres and thicknesses of some 10 nm. They were usually connected with t-Al_3_X particles that, in addition to Zr, also contained Sc, Y, and Si, and served as effective heterogeneous nucleation sites. The number density of t-Al_3_X particles was much smaller in 6086-Sc and 6086-ScY than in the reference alloy. This effect was more critical at the centres, where the supersaturation with Mn and Si is less due to their partition coefficient that is smaller than one. Thus, only a few α-AlMnSi particles formed during heating can grow to immense dimensions. The supersaturation with Mn and Si was higher close to the interdendritic regions; thus, many particles can form.

Many spherical particles in the region with larger plates were identified as L1_2_-Al_3_X precipitates using EDS mapping (Figure 7 and Figure 8) and HRTEM micrographs (Figure 9). In each case, they possessed a spherical morphology, diameter between 30 nm and 40 nm, and cube-to-cube orientation relationship with the α-matrix. The size of L1_2_ precipitates was 36.1 ± 7.2 nm in 6086-Sc and 33.1 ± 5.6 nm in 6086-ScY, which can be considered almost the same.

EDS showed that L1_2_ precipitates contain Si in every condition. Dorin et al. argued [61] that Si is essential during clustering, but was expelled from the particles when stabilising into the L1_2_ structure. The particles contained Y in the Y-modified alloy. Plate-like AlX_2_Si_2_ particles were rarely observed in 6086-Sc and 6086-ScY, where X stands for Sc, Zr, and Y (Figure 6d).

### 3.2. Heat Treatment of the Homogenized Alloys

Heat treatment was carried out at temperatures above the second peak on the DSC curves (Figure 4a). Thus, holding the alloys at 350 °C, 400 °C, and 450 °C, respectively, up to 5 h did not cause any substantial microstructure changes. XRD patterns show that the positions, shapes, and sizes of the diffraction peaks of the homogenized and isothermally heat-treated samples virtually did not change (Figure 10). The results of DSC show rather sluggish reaction in the endothermic region (Figure 11a), and a very low change in dimensions (Figure 11b). This effect can be explained by the fact that almost complete precipitation of dispersoids occurred during heating and homogenization, and only minor changes happened during further annealing of these alloys.

The microstructures also remained the same (Figure 12). At the dendrite centres, a few micrometres long and very thin α-AlMnSi plates were present, with plenty of spherical L1_2_-Al_3_X precipitates. Their structure and composition remained the same as after homogenization (Figure 9).

The elemental mapping in other regions (Figure 13 and Figure 14) showed that some of the finest plates could be attributed to the AlSc_2_Si_2_ phase, which also incorporates some Zr and Y, and therefore, is detrimental to mechanical properties, first due to its shape, which may promote brittleness, and second because it leaves less Sc, Zr, and Y for the formation of desirable L1_2_-precipitates.

The size of L1_2_ precipitates that formed during homogenization only slightly increased with further heat treatments. The precipitate sizes were 37.3 ± 4.9 nm for 6086-Sc and 35.2 ± 3.9 nm for 6086-ScY. Higher magnification micrographs showed that new smaller precipitates appeared (5.2 ± 3.1 nm in 6086-Sc and 3.9 ± 2.6 nm in 6086-ScY) in addition to already present precipitates (Figure 15). These smaller precipitates appeared in lines. It seems that they nucleated at the dislocations or in their near vicinity.

A relatively small decrease in hardness appeared after homogenization (Figure 16). This effect can be explained by decreased solid solution hardening and dispersoid coarsening. The highest hardness drop occurred during the first hour of holding, and after that, it remained almost the same. The 6086-ScY alloy had the lowest initial hardness, but this hardness did not change with time at 350 °C, and was finally the highest among all alloys and conditions. The hardness of these alloys in the T6 temper was about 120 HV. It means that the alloys are soft after homogenization and further isothermal treatment. The presence of dispersoids may increase the thermal stability of the alloys, which should be proved by further investigation.

### 3.3. Heat Treatment of the Alloys in the As-Cast Condition

The alloys in the as-cast condition were heat-treated in the temperature range between 200 °C and 450 °C for up to 5 h. This temperature range encompasses the second peak on the DSC curves (Figure 4a) and the third peak over its maximum. Holding the alloys for 5 h at 400 °C caused the dissolution of Θ-Al_2_Cu, while peaks of the other phases remained unchanged (Figure 17).

Figure 18 shows the microstructures of the samples heat-treated for 5 h at 300 °C and 450 °C, respectively. The primary phases remained unchanged at 300 °C, while the growth and spheroidization of the primary Mg_2_Si took place at 450 °C, as also found in other investigations [62]. No precipitates or dispersoids were visible in the α-Al matrix at 300 °C. On the other hand, dispersoids were visible in α-Al of all samples heat-treated at 450 °C. Their distribution was much more uniform at dendrite centres and interdendritic regions than in the homogenized samples.

The observation of alloys heat-treated at 300 °C revealed precipitates in the α-Al at higher magnifications (Figure 19 and Figure 20). There were present at least two types of precipitates. The darker plates in the matrix were β′-precipitates, while the brighter plates were Q’-AlCuMgSi, and, in some cases, they could also be Θ′-Al_2_Cu. Larger particles were present in the interdendritic regions, while they were hardly visible at the centres. The number density of β′-precipitates decreased at 350 °C, and they were not present in samples heat-treated at 400 °C and 450 °C.

Holding at 300 °C and 400 °C caused an exothermal effect (Figure 21). The exothermal effects of the reference alloy 6086 and 6086-ScY were of similar magnitudes at both temperatures; however, the thermal effect in 6086 was much stronger. The thermal effect of the 6086-Sc was slightly lower than that of the reference alloy at 300 °C, but much stronger at 400 °C. The behaviour of 6086-Sc was similar in dilatometric tests (Figure 22).

Figure 23 shows the effect of T5 heat treatment on the room temperature hardness of the investigated alloys. The hardness of the reference alloys decreased sharply up to 350 °C, then started to rise, probably due to the formation of dispersoids. The hardness of 6086-Sc and 6086-ScY decreased to 300 °C, where the hardness was the same as in the reference alloy. The hardness of 6086-Sc and 6086-ScY was much higher than that of the reference alloy between 350 °C and 450 °C. This effect can be attributed to the formation of L1_2_-precipitates. After homogenization, the hardness of all alloys was the same. Generally, the additions of Sc and Y make the alloy much more heat resistant in the temperature range between 350 °C and 450 °C.

The hardness was lower after T5 treatment than in the as-cast state. The most significant drop occurred after the first hour of treatment, while prolonged treatment did not change the hardness significantly. Nevertheless, the hardness was substantially higher than after homogenization and additional treatment.

### 3.4. Compression Tests of the Specimens after Dilatometry

Table 2 and Figure 24 give the results of compression test carried out on the specimen tested with dilatometry. The maximum true stress was achieved at 6086-Sc and 6086-ScY isothermal annealing of the cast specimens at 400 °C, which was also consistent with the hardness. The maximum true stress for the homogenized and isothermally treated samples was about 20–50 MPa lower.

The type of treatment strongly affected the true stress–true strain deformation curves. The strain hardening took place up to the end of the tests in homogenized and isothermally treated samples (Figure 24a). On the other hand, the curves of the isothermally treated cast specimens exhibited high strain hardening up to 0.15–0.20 true strain and thereafter, strain softening occurred. The reason is not clear; however, it may be related to the type of dispersoids present in each sample. In homogenized samples prevailed incoherent α-AlMnSi dispersoids, which induce strong work hardening, on the other hand, isothermally treated cast samples coherent dispersoids prevail, which can lead to shearing. This topic is going to be investigated in future research.

## 4. Discussion

In our previous work, we calculated equilibrium phases in AA 6086 [51] and AA 6086 with 0.2 wt.% Sc [55] (Thermo-Calc, database TCAl5). This database does not contain information about yttrium phases; therefore, we were not able to perform calculations for Y-modified alloy. Nevertheless, the addition of Y is relatively small, only minor changes are expected in comparison to the Sc modified 6086 [63]. It was calculated that in the investigated temperature range, the following phases, which may form dispersoids, are thermodynamically stable: α-AlMnSi, Si_2_Zr, AlSc_2_Si_2_, and Al_13_Cr_4_Si_4_, while both t-Al_3_Zr and L1_2_-Al_3_Sc are metastable. Si_2_Zr and Al_13_Cr_4_Si_4_ were not found as individual particles. Si_2_Zr was mainly attached to metastable t-Al_3_Zr. Chromium was found in α-AlMnSi, and enrichment of this element in α-AlMnSi was sometimes observed (Figure 7). AlSc_2_Si_2_ usually appears in a plate-like shape and is detrimental to the mechanical properties of the alloys. Both Zr and Y dissolve in this phase in our alloys, thus forming Al_2_(Sc, Zr, Y)_2_Si_2_, which is briefly indicated as AlX_2_Si_2_. The same was found for t-Al_3_Zr and L1_2_-Al_3_Sc; therefore, it is better to write t-Al_3_X and L1_2_-Al_3_X, the dispersoids can have variable composition.

The primary α-AlMnSi formed during solidification and was present as large particles in the interdendritic regions and grain boundaries. It hardly changed during subsequent heat treatments. It is also crucial to prevent excessive grain growth [43]. Much more important are α-AlMnSi dispersoids. Our results suggest that they mainly form in the temperature range of the third peak between 380 °C and 490 °C (Figure 4) and that they only coarsen during homogenization. They can form on dislocations and β′-precipitates [64]. Their nucleation can be facilitated by the additions of Cd [48] and Mo [49]. In alloy 6086, many α-AlMnSi dispersoids heterogeneously nucleated on t-Al_3_Zr [51]. This effect can contribute to a much finer distribution of α-AlMnSi dispersoids in the reference alloy after homogenization and can lead to synergistic strengthening by nano-sized α-AlMnSi and t-Al_3_Zr [65]. The additions of Sc and (Sc + Y) caused the formation of a small number of huge α-AlMnSi plates, especially at dendrite centres, where is the lowest Mn content due its segregation in the interdendritic region. In this region, a larger fraction of L1_2_-Al_3_X dispersoids was formed. Since Zr incorporates into L1_2_-Al_3_X, its supersaturation decreases and cannot form many t-Al_3_X particles. Thus, only a few α-AlMnSi dispersoids form, which can grow to large dimensions. A similar phenomenon was found by De Luca et al. [50]. However, in their case, α-AlMnSi dispersoids were predominantly present in dendrite centres. The non-uniform distribution of α-AlMnSi occurred during conventional homogenization, when the Sc and (Sc + Y) alloys were heated to 520 °C. However, when the alloys were isothermally heat treated from the as-cast state (T5 temper), they were many and were rather uniformly distributed α-AlMnSi dispersoids formed at 400 °C. At this temperature, Mn diffusion is more sluggish, and the supersaturation is higher than at 520 °C. As a result, the combination of a higher driving force and lower diffusivity stimulates higher nucleation rate of α-AlMnSi everywhere. The same phenomenon was also found previously. Rakhmonov et al. [66] and Qian et al. [67] found a beneficial effect of a low-temperature homogenization on the properties of AA 6082 caused by the formation of a higher density of small α-AlMnSi dispersoids.

As mentioned before, L1_2_-precipitates formed mainly at dendrite centres, where only a few α-AlMnSi were present. However, in the regions with a high number density of α-AlMnSi, L1_2_ was relatively rare. The reason is not apparent. Nevertheless, the silicon in the solid solution can have an essential role since Si strongly accelerates the nucleation of L1_2_-Al_3_X precipitates [25]. The formation of α-AlMnSi dispersoids can reduce the Si in the solid solution, making the nucleation of L1_2_-precipitates harder, even though the content of Sc was higher in the interdendritic regions than in dendrite cores. DSC and dilatometry results suggest that α-AlMnSi dispersoids form first and predominantly during heating to the homogenization temperature, while most L1_2_-precipitates forms later during heating and holding at the homogenization temperature. The precipitates are relatively coarse with larger interparticle distance, weakly contributing to Orowan strengthening. Further heat treatment of the homogenized samples did not cause any significant effect on the microstructure and hardness, which DSC and dilatometry also confirmed. Homogenization treatment produced rather large L1_2_-Al_3_X precipitates, containing a lot of Sc, Zr, and Y. These elements become unavailable for precipitation during further isothermal heat treatment, and only a small fraction of new nanosized L1_2_ dispersoids can form, which cannot substantially increase the hardness.

It was evident that the Al-rich matrix was supersaturated with alloying elements after casting, which allowed the formation of dispersoids by the so-called T5 treatment, ageing of the alloys in the as-cast condition. The advanced Al-Mg-Si alloys are multicomponent and multiphase. Therefore, several interdependent processes can occur by any treatment, and composition modification can have different effects. The addition of Sc and (Sc + Y) caused the formation of L1_2_ precipitates, which is the main reason for much better properties in the temperature range 350–450 °C, where the reference alloy rapidly softens. The α-AlMnSi dispersoids still predominantly formed close to the interdendritic regions, but large α-AlMnSi dispersoids were prevented at the dendrite centres. Precipitation of L1_2_-Al_3_X is possible by adding Sc, but a small addition of Y makes the processes more sluggish and can improve high-temperature strength. The hardness attains approximately 70% of the maximum hardness of the reference alloy in T6 temper. We believe that there is room for optimizing heat treatment to obtain a good combination of room temperature strength and thermal resistance.

## 5. Conclusions

The main conclusions of this work are based on the microstructural characterisation and the results of differential scanning calorimetry and dilatometry:

The additions of Sc and combinations of Sc and Y caused the formation of large α-AlMnSi plates and plenty of nanometre-sized spherical L1_2_ precipitates during homogenization at dendrite centres, while smaller α-AlMnSi and t-Al_3_Zr formed elsewhere.

A small addition of Y did not induce the formation of any new yttrium-rich phases. Y was mainly incorporated in other phases already present in 6086 with the addition of Sc: AlSc_2_Si_2_, t-Al_3_Zr, and L1_2_-Al_3_Sc.

Dilatometry and DSC during continuous and isothermal treatments give an insight into the intensity of processes. Adding Y slightly decreases the kinetics of the reactions and sizes of L1_2_-precipitates.

T5 treatment (heat treatment of the as-cast alloys) was superior in the uniformity of dispersoids and attained mechanical properties compared to the T6 treatment (homogenization followed by artificial ageing).

The results provide a sound basis for optimising heat treatment of Al-Mg-Si alloys microalloyed with Zr, Sc, and or Y. The classical homogenization treatment may not be the best option.

The results also provide a starting point for studying the thermal stability of the investigated alloys for possible applications at higher temperatures.

## Figures and Tables

**Figure 1 materials-16-02949-f001:**
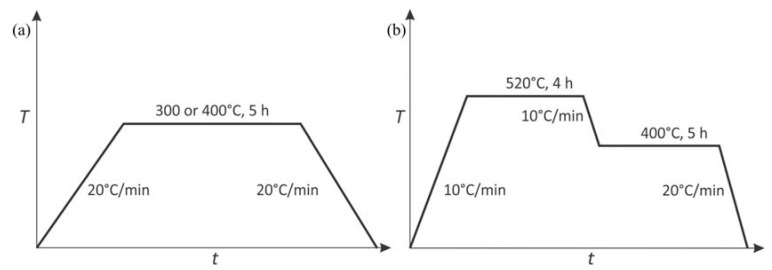
Time-temperature diagrams for following the processes during dilatometry tests and differential scanning calorimetry: (**a**) isothermal annealing at 300 °C and 400 °C, (**b**) processes during homogenization and subsequent holding at 400 °C.

**Figure 2 materials-16-02949-f002:**
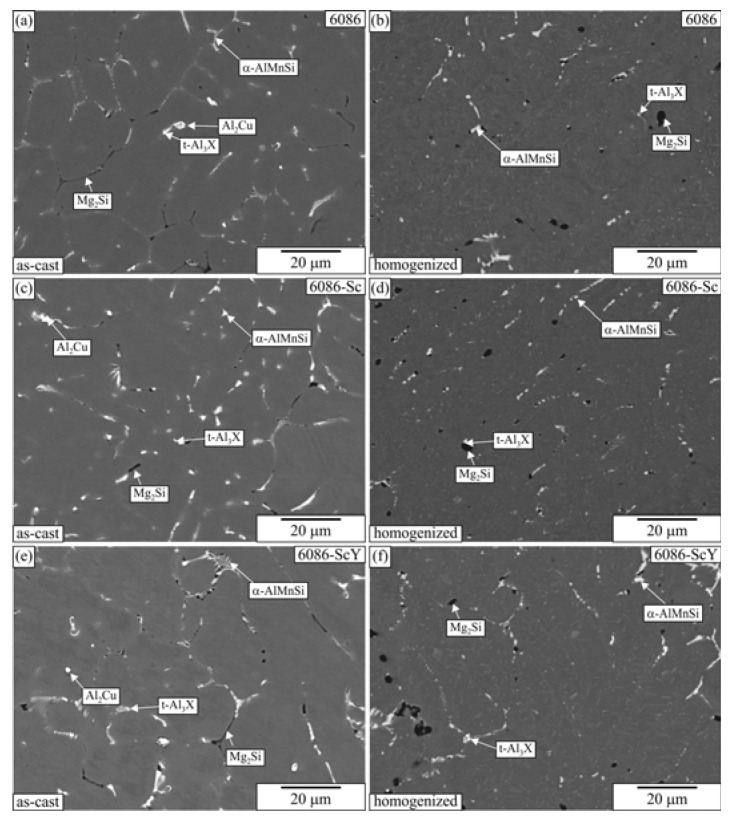
Microstructures of the investigated alloys in the as-cast and homogenized conditions (backscattered electron micrographs, sample diameter 10 mm). (**a**,**b**) Alloy 6086, (**c**,**d**) 6086-Sc, (**e**,**f**) 6086-ScY.

**Figure 3 materials-16-02949-f003:**
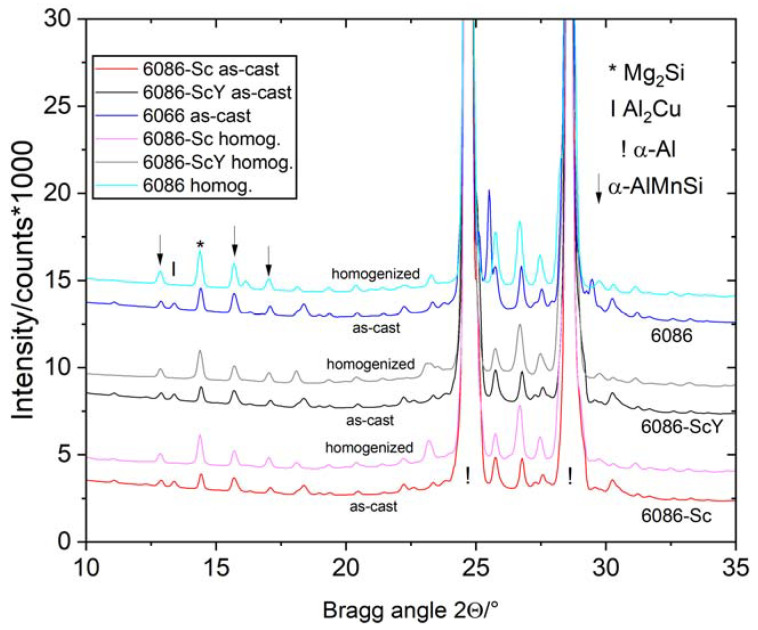
XRD patterns of the investigated alloys in the as-cast and homogenized conditions. The most important peaks of the identified phases are indicated.

**Figure 4 materials-16-02949-f004:**
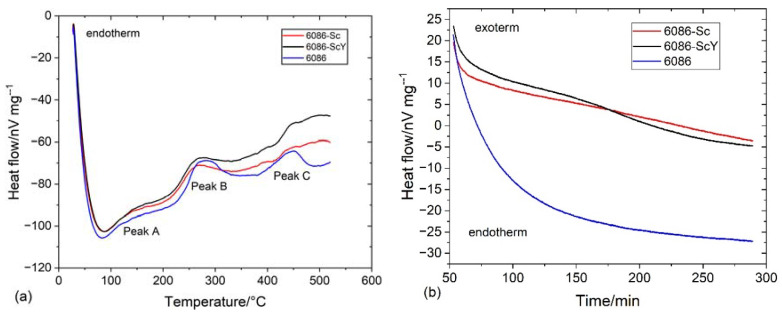
Heat flow by DSC tests of the investigated alloys (**a**) during heating from RT to 520 °C with 10 °C/min, and (**b**) during holding at 520 °C for 4 h.

**Figure 5 materials-16-02949-f005:**
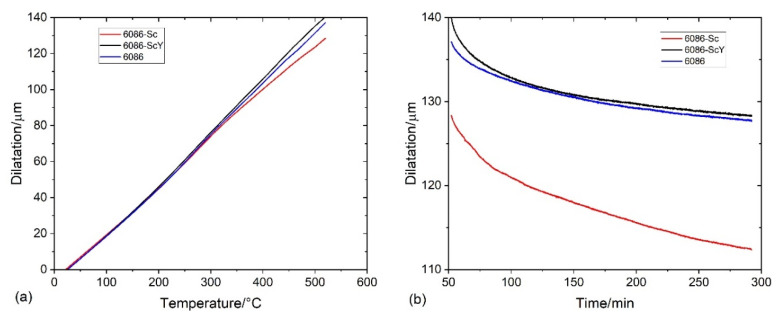
Dilatation of the investigated alloys (**a**) during heating from RT to 520 °C with 10 °C/min, and (**b**) during holding at 520 °C for 4 h.

**Figure 6 materials-16-02949-f006:**
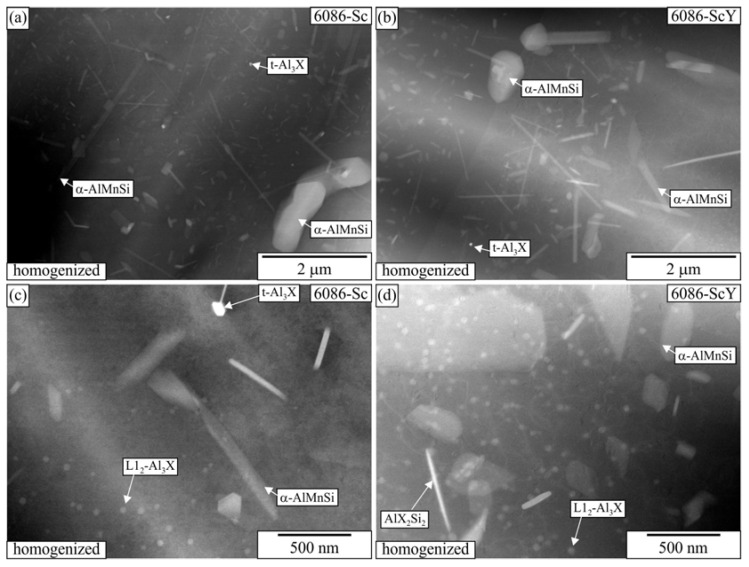
HAADF-TEM micrographs of the sample 6086–Sc and 6086_ScY after homogenization (520 °C, 6 h). A larger area at a grain boundary (**a**) 6086-Sc, (**b**) 6086-ScY. An area at the dendrite centre, (**c**) 6086-Sc, and (**d**) 6086-ScY.

**Figure 7 materials-16-02949-f007:**
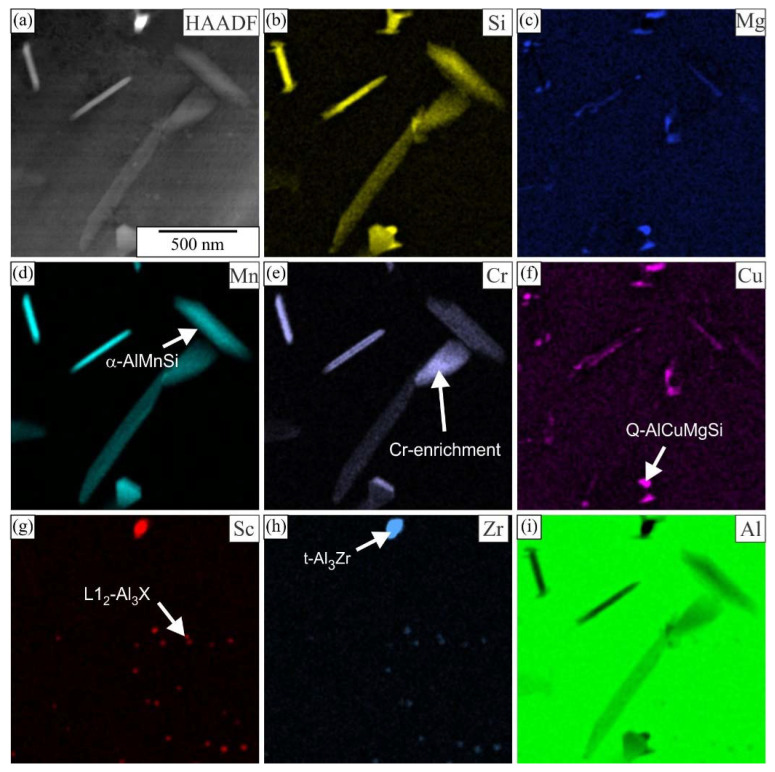
Distribution of the elements within the matrix of the alloy 6086-Sc after homogenization (**a**) HAADF micrograph, EDS mapping of selected elements, (**b**) Si, (**c**) Mg, (**d**) Mn, (**e**) Cr, (**f**) Cu, (**g**) Sc, (**h**) Zr, and (**i**) Al.

**Figure 8 materials-16-02949-f008:**
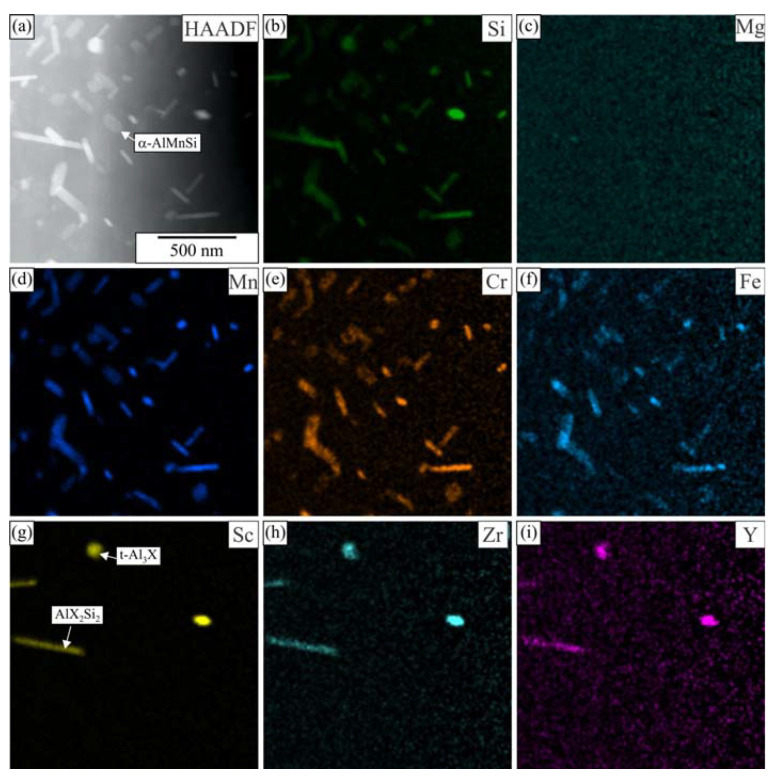
Distribution of the elements within the matrix of the alloy 6086-ScY after homogenization (**a**) HAADF micrograph, EDS mapping of selected elements, (**b**) Si, (**c**) Mg, (**d**) Mn, (**e**) Cr, (**f**) Fe, (**g**) Sc, (**h**) Zr, and (**i**) Y.

**Figure 9 materials-16-02949-f009:**
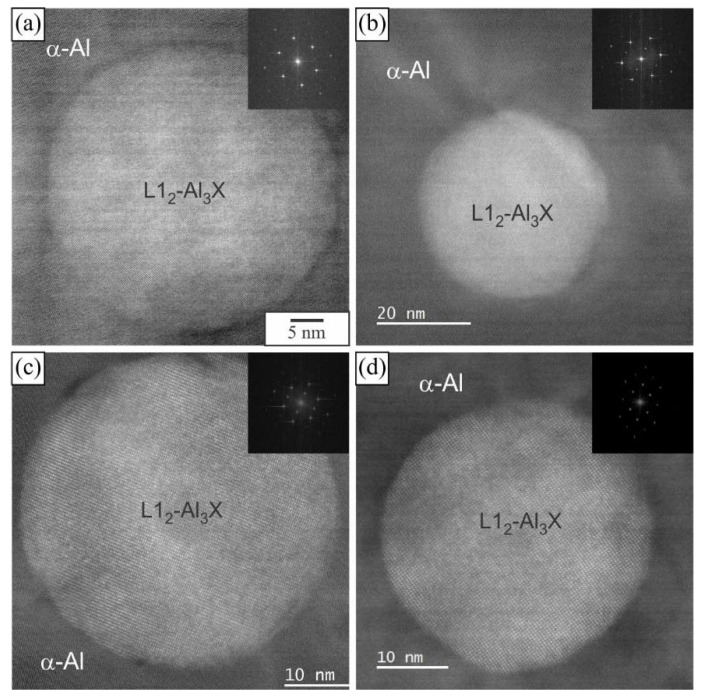
The results of analytical TEM of a particle with the L1_2_ structure. After homogenization: (**a**) 6086-Sc, (**b**) 6086-ScY; and after homogenization and heat treatment at 400 °C for 5 h: (**c**) 6086-Sc and (**d**) 6086-ScY.

**Figure 10 materials-16-02949-f010:**
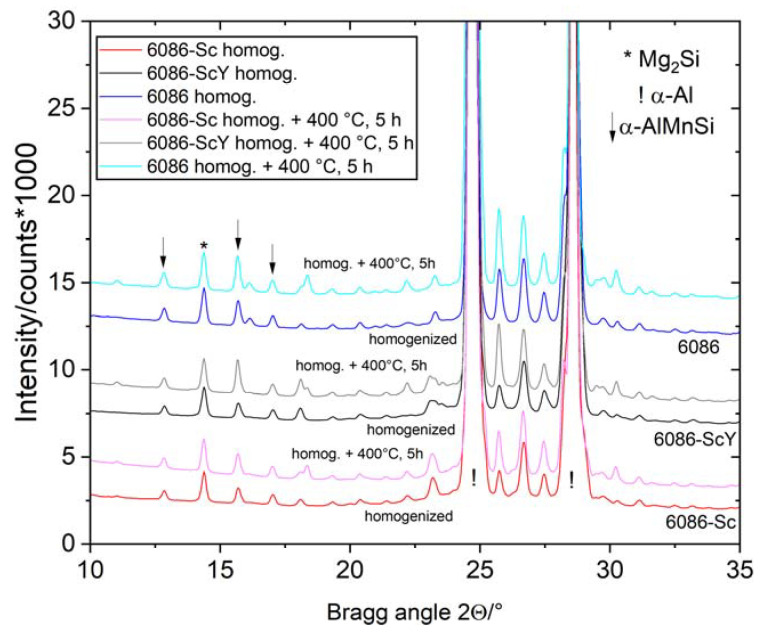
The XRD patterns of the investigated alloys in the homogenized condition (520 °C, 5 h) and isothermally heat-treated at 400 °C for 5 h. The most important peaks of the identified phases are indicated.

**Figure 11 materials-16-02949-f011:**
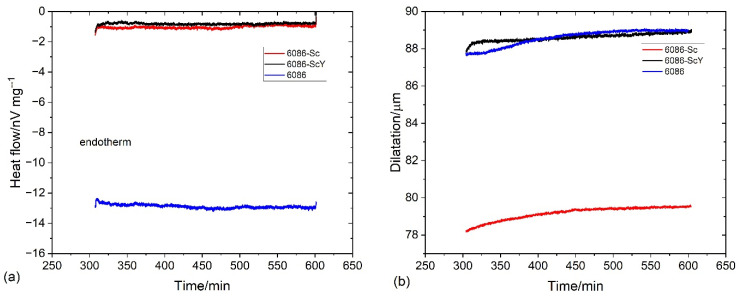
The thermal effect and dilatation of the homogenized samples during holding at 400 °C for five hours. (**a**) Heat flow; (**b**) dilatation.

**Figure 12 materials-16-02949-f012:**
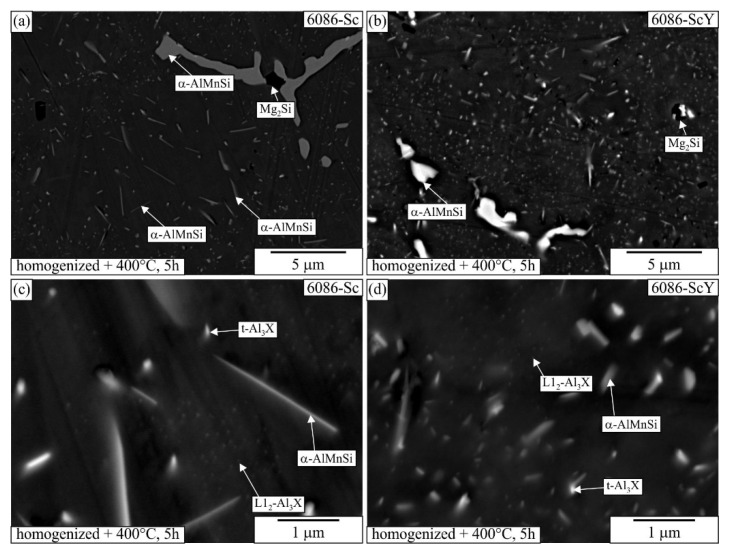
Backscattered electron micrographs of the alloys (**a**,**c**) 6086-Sc and (**b**,**d**) 6086-ScY after homogenization and holding for 5 h at 400 °C.

**Figure 13 materials-16-02949-f013:**
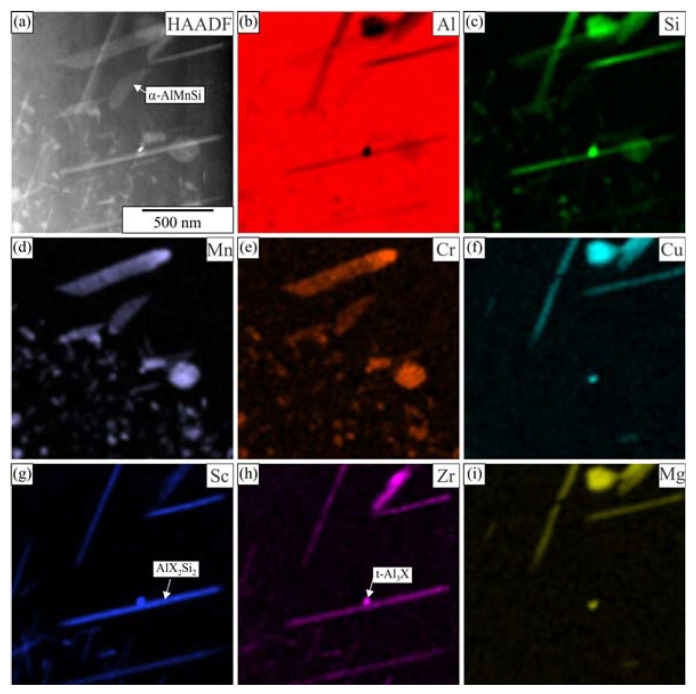
Distribution of the elements within the matrix of the alloy 6086-Sc after homogenization and holding for 5 h at 400 °C (**a**) HAADF micrograph, EDS mapping of selected elements (**b**) Al, (**c**) Si, (**d**) Mn, (**e**) Cr, (**f**) Cu, (**g**) Sc, (**h**) Zr, and (**i**) Mg.

**Figure 14 materials-16-02949-f014:**
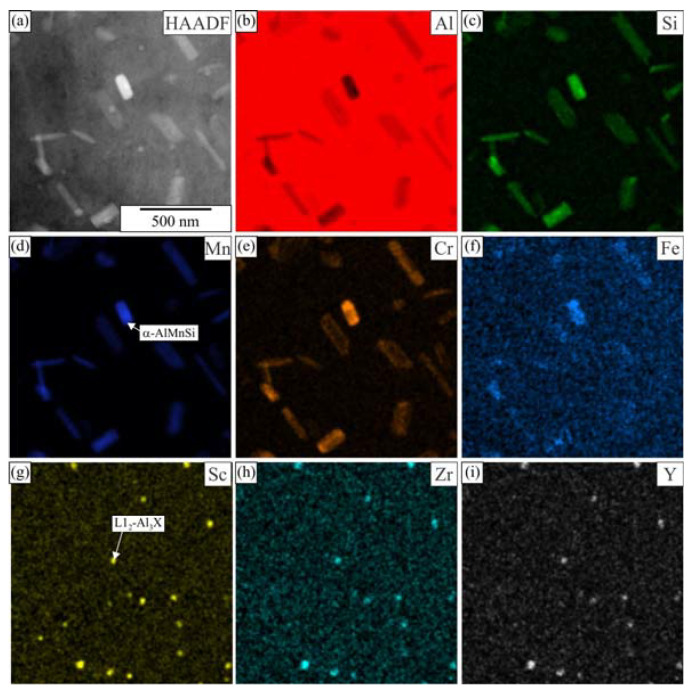
Distribution of the elements within the matrix of the alloy 6086-ScY after homogenization and holding for 5 h at 400 °C. (**a**) HAADF micrograph, EDS mapping of selected elements (**b**) Al, (**c**) Si, (**d**) Mn, (**e**) Cr, (**f**) Fe, (**g**) Sc, (**h**) Zr, and (**i**) Y.

**Figure 15 materials-16-02949-f015:**
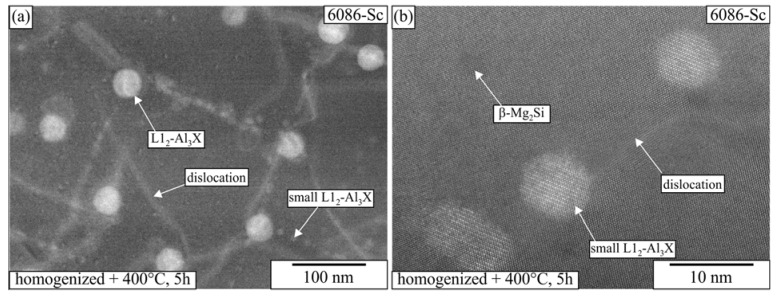
HAADF micrograph of L1_2_-Al_3_X dispersoids in 6086-Sc after homogenization and isothermal treatment at 400 °C for 5 h. (**a**) A lower magnification with large and small dispersoids, (**b**) A higher magnification with some smaller dispersoids.

**Figure 16 materials-16-02949-f016:**
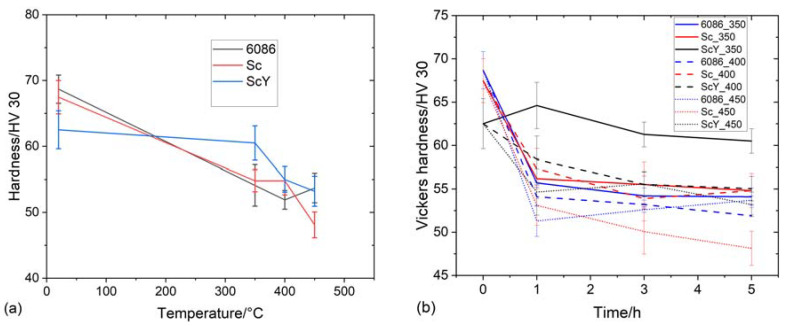
Heat treatment of the alloys in the homogenized condition. (**a**) The effect of the five-hour isochronal ageing on the hardness of the investigated alloys at different temperatures. (**b**) The effect of time on hardness at isothermal treatment at 350 °C, 400 °C, and 450 °C.

**Figure 17 materials-16-02949-f017:**
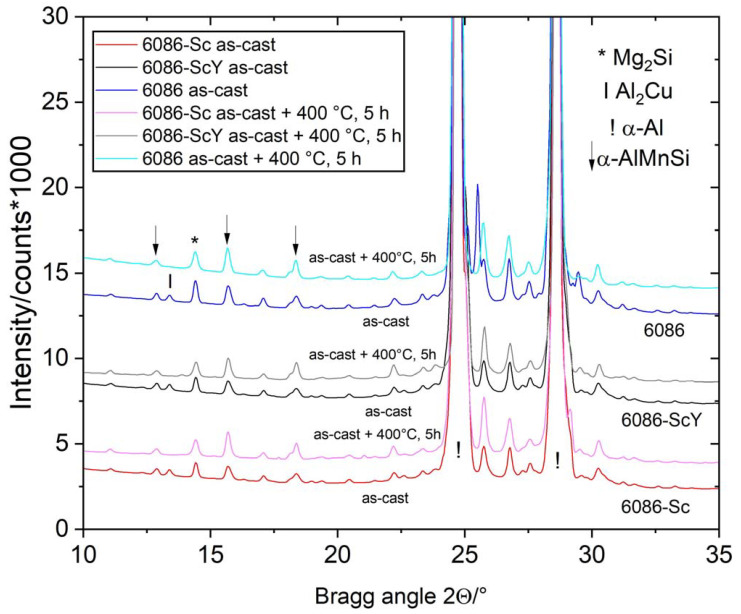
XRD patterns of the investigated alloys in the as-cast condition and after holding for 5 h at 400 °C.

**Figure 18 materials-16-02949-f018:**
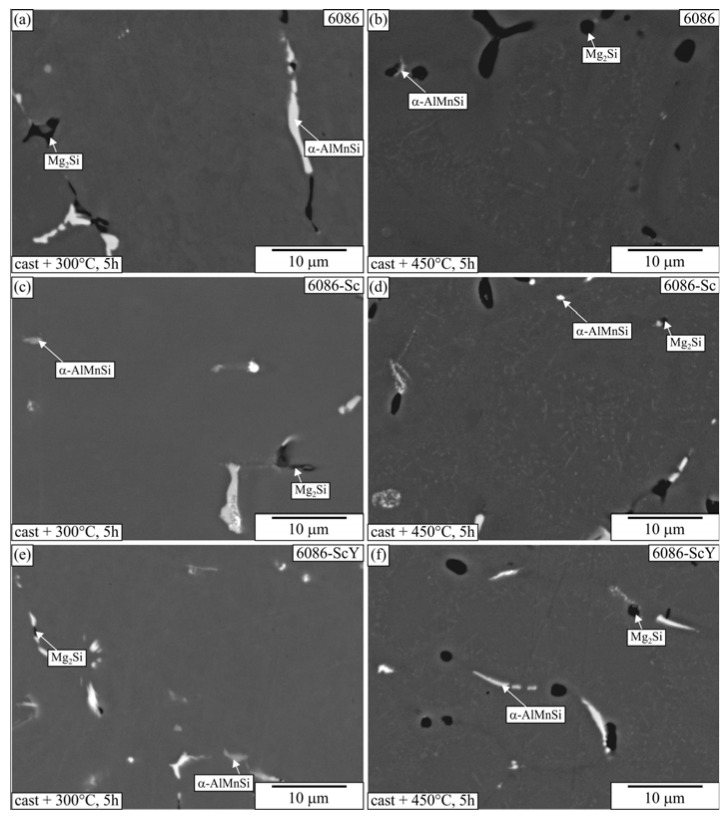
Microstructures of the investigated alloys after T5 heat treatment of the as-cast samples for 5 h at 300 and 450 °C (backscattered electron micrographs, sample diameter 10 mm). (**a**) Alloy 6086, 300 °C, (**b**) 6086, 450 °C, (**c**) 6086-Sc, 300 °C, (**d**) Alloy 6086-Sc, 450 °C, (**e**) 6086-ScY, 300 °C, (**f**) 6086-ScY, 450 °C.

**Figure 19 materials-16-02949-f019:**
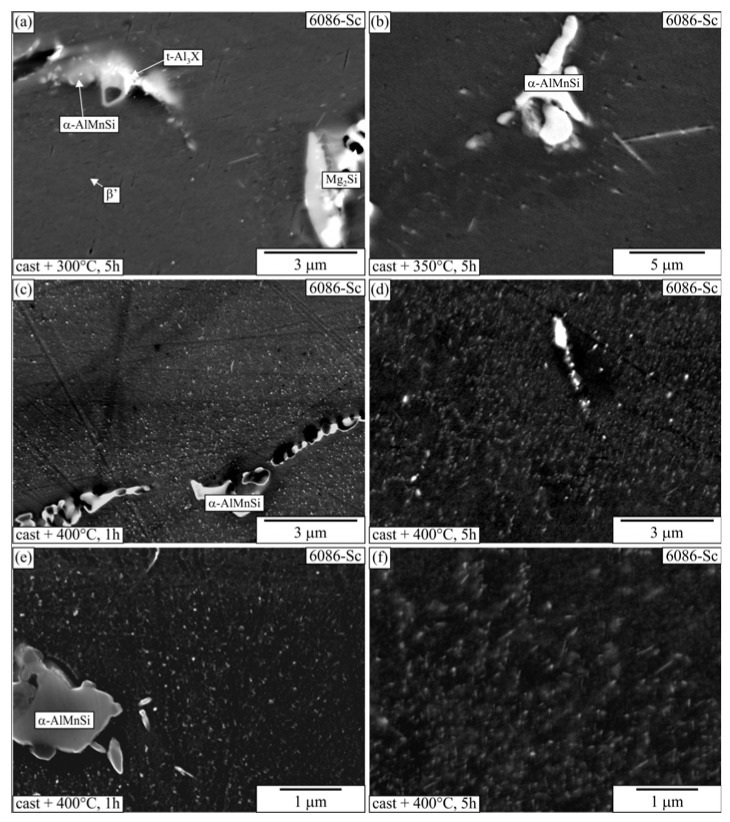
Microstructures of the 6086-Sc after T5 heat treatment of the as-cast samples at different temperatures and durations (backscattered electron micrographs, sample diameter 10 mm). (**a**) 300 °C, 5 h, (**b**) 350 °C, 5 h, (**c**) 400 °C, 1 h, (**d**) 400 °C, 5 h, (**e**) 400 °C, 1 h, (**f**) 400 °C, 5 h.

**Figure 20 materials-16-02949-f020:**
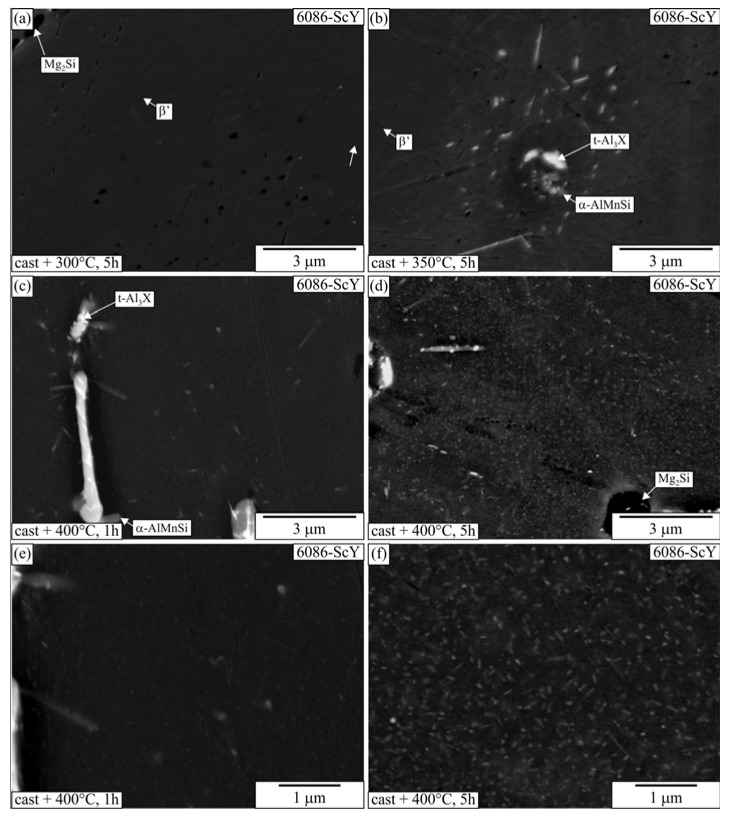
Microstructures of the 6086-ScY after T5 heat treatment of the as-cast samples at different temperatures and durations (backscattered electron micrographs, sample diameter 10 mm). (**a**) 300 °C, 5 h, (**b**) 350 °C, 5 h, (**c**) 400 °C, 1 h, (**d**) 400 °C, 5 h, (**e**) 400 °C, 1 h, (**f**) 400 °C, 5 h.

**Figure 21 materials-16-02949-f021:**
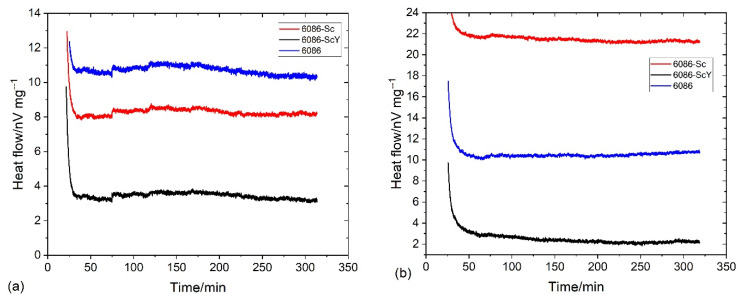
The results of DSC during isothermal annealing (**a**) at 300 °C, (**b**) at 400 °C.

**Figure 22 materials-16-02949-f022:**
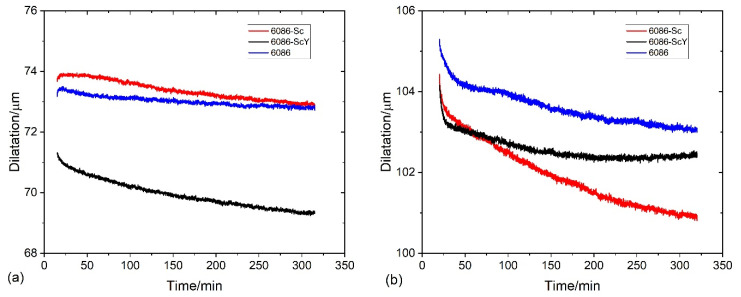
The results of the dilatometry during isothermal annealing (**a**) at 300 °C, (**b**) at 400 °C.

**Figure 23 materials-16-02949-f023:**
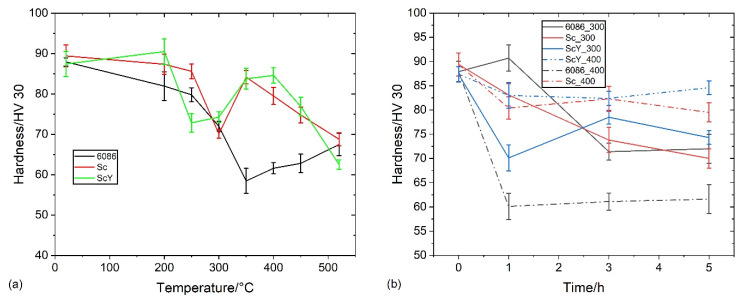
Heat treatment of the investigated alloys in the as-cast condition. (**a**) The effect of the five-hour isochronal ageing on the hardness at different temperatures. (**b**) The effect of the isothermal treatments duration on hardness at 300 °C and 400 °C.

**Figure 24 materials-16-02949-f024:**
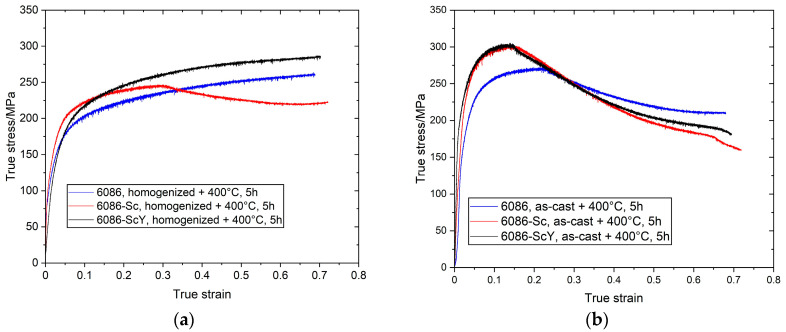
True stress—true strain diagrams of samples exposed to dilatometry. (**a**) After sequence simulating homogenization and isothermal heat treatment at 400 °C for 5 h; (**b**) after sequence simulating isothermal treatment of the as-cast specimens at 400 °C for 5 h.

**Table 1 materials-16-02949-t001:** The chemical compositions of the investigated alloys in wt.% as determined using AES-ICP (Atomic Emission Spectroscopy—Inductively Coupled Plasma, Agilent 5800 VDV (Vertical Dual View), Agilent Technologies Inc., Santa Clara, CA, USA).

Alloy	Si	Fe	Cu	Mn	Mg	Cr	Zn	Ti	Zr	Sc	Y	Al
6086	1.58	0.18	0.55	0.70	1.00	0.19	0.02	0.04	0.17	-	-	balance
6086-Sc	1.37	0.17	0.49	0.72	0.88	0.17	0.02	0.04	0.16	0.22	-	balance
6086-ScY	1.36	0.17	0.49	0.72	0.89	0.17	0.02	0.04	0.16	0.18	0.09	balance

**Table 2 materials-16-02949-t002:** Results of the compression tests.

Alloy	Maximum True Stress MPa	True Strain at Maximum True Stress
6086, homogenized + 400 °C, 5 h	263.6	0.684
6086-Sc, homogenized + 400 °C, 5 h	247.1	0.301
6086-ScY, homogenized + 400 °C, 5 h	287.2	0.693
6086, as-cast + 300 °C, 5 h	266.1	0.143
6086-Sc, as-cast + 300 °C, 5 h	282.7	0.151
6086-ScY, as-cast + 300 °C, 5 h	298.9	0.122
6086, as-cast + 400 °C, 5 h	272.3	0.197
6086-Sc, as-cast + 400 °C, 5 h	302.2	0.141
6086-ScY, as-cast + 400 °C, 5 h	305.2	0.138

## Data Availability

The data presented in this study are available on request from the corresponding author.

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
