# Peer review of "Dispersoids in Al-Mg-Si Alloy AA 6086 Modified by Sc and Y"

_materials, 2023, doi:10.3390/ma16082949_

Round 1

Reviewer 1 Report

The article entitled “ Effect of Sc and Y on the formation of dispersoids in AA 6086” authored by Zupanic et al. reports the effect of isothermal heat treatment on the microstructure and hardness of the Al alloys. The article has 24 figures, 1 table and 42 references. The qualities of figures are good barring few. The authors established their claims with the aid of SEM, TEM, EDS, XRD and DSC. The references are recent and relevant to the chosen study. Though the authors are appreciated for undertaking an experimental study, the following queries needs to be addressed before considering for publication.

1.      Authors must expand the Acronyms used in the manuscript at the first instant of usage.

2.      Title does not convey the work carried out in the present study. It is recommended to modify the title.

3.      The present work seems to be a continuation of a previous study [Ref.33]. The effects of Sc addition are already reported on the article.

4.      More emphasis on presenting the novelty of the work is recommended.

5.      Authors need to state how the isothermal temperature and duration were fixed. If there is any global standard available the same may be included.

6.      Inclusion of a nomenclature section is recommended. Page 2 line 77- What is Yl50?

7.      Careless mistakes to be pruned. Ex. Page 2 line 51 and 52- Statement to be modified. Line 67,68- Check the font used.

8.      Why 3 different alloys were chosen. More substantiation is warrented.

9.      Author needs to used the terms used by researchers worldwide should be used. Ex Lobotom 5 and Isomet 4000?

10.  Features in the microstructure should be marked on the images itself.

11.  It is recommended to add few mechanical tests.

12.  Authors need to compare their results with published literature. The impact of isothermal heat treatment on the microstructure needs more substantiate discussion.

13.  The duration of hardness needs to be supplied.

Author Response

We thank the reviewer for the careful review of the article. We tried to respond as best as possible to the remarks and suggestions.

  1. Authors must expand the Acronyms used in the manuscript at the first instant of usage.

We have checked all acronyms, and they are expended.

  1. Title does not convey the work carried out in the present study. It is recommended to modify the title.

We believe that our work is devoted to dispersoids in the investigated alloys. Thus we changed the title to: “Dispersoids in Al-Mg-Si alloy AA 6086 modified by Sc and Y”

  1. The present work seems to be a continuation of a previous study [Ref.33]. The effects of Sc addition are already reported on the article.

The previous work only slightly overlaps with the current one. All isothermal treatments, compression tests, dilatometry and DSC, is new research.

  1. More emphasis on presenting the novelty of the work is recommended.

We tried to present the novelty more clearly. It is written in the Introduction, lines 105-114.

  1. Authors need to state how the isothermal temperature and duration were fixed. If there is any global standard available the same may be included.

There is no global standard. We studied the literature in the same area, and in addition, we carried out some preliminary experiments to obtain a good enough starting point.

  1. Inclusion of a nomenclature section is recommended. Page 2 line 77- What is Yl50?

We simplified the notation of samples and experiments. “Y150” was a lapsus that was corrected.

  1. Careless mistakes to be pruned. Ex. Page 2 line 51 and 52- Statement to be modified. Line 67,68- Check the font used.

Thank you for the remark. We check the whole text.

  1. Why 3 different alloys were chosen. More substantiation is warrented.

Thank you for the comment. It is explained in more detail in Materials and Methods, lines 120-125.

  1. Author needs to use the terms used by researchers worldwide should be used. Ex Lobotom 5 and Isomet 4000?

We checked the names of our equipment: The alloys were sectioned using two metallographic saws, Labotom 5 (Struers, Ballerup, Denmark) and IsoMet 1000 (Buehler, Lake Bluff, Illinois, USA).

  1. Features in the microstructure should be marked on the images itself.

We have marked features on the micrographs.

  1. It is recommended to add few mechanical tests.

In this short period, we were able to perform compression tests using samples tested by dilatometry.

  1. Authors need to compare their results with published literature. The impact of isothermal heat treatment on the microstructure needs more substantiated discussion.

We added a new subsection “Discussion” to provide a more detailed discussion of the results.

  1. The duration of hardness needs to be supplied.

Further details are given in Materials and Methods, lines 177-178

Reviewer 2 Report

This manuscript investigates the addition and Sc and Y into alloy AA 6068 on modifying the microstructure, phase present and hardness using multiple characterization techniques. There are a number of concerns that reviewer suggests to address:

1. Materials and methodology sections need to be better organized. In particular, it is not clear that the heat treatment is conducted in the same facility as the dilatometry and DSC and whether the protocol in figure 1 is also applied as that for heat treatment.

2. Enough support or clarification need to be present in the results analysis section. For example, in the discussion "The brightest particles are tetragonal Al3Zr (t-Al3Zr), which in 6086-Sc and 6086-ScY alloys contained these two microalloying elements as well. In addition to these phases, small amounts of minor phases, such as theta-Al2Cu, Q-AlCuMgSi, AlSc2Si2 and ZrSi2, were also identified".  How those phases are identified need to be provided and they should be marked in the figure for direct identification.

3. A discussion section needs to be provided before the conclusion, and overall presentation needs to be improved.

4. The English needs to be checked throughout the manuscript. Multiple grammar errors were caught, for example, "The effect of yttrium on the properties of Al-alloys was much less investigated that 65 the effects of Sc and Zr".

Author Response

We thank the reviewer for the careful review of the article. We tried to respond as best as possible to the remarks and suggestions.

  1. Materials and methodology sections need to be better organized. In particular, it is not clear that the heat treatment is conducted in the same facility as the dilatometry and DSC and whether the protocol in figure 1 is also applied as that for heat treatment.

Thank you for the suggestion. We explained the difference between heat treatment and protocol for DSC and dilatometry (Materials and methods, Lines 166-172)

  1. Enough support or clarification need to be present in the results analysis section. For example, in the discussion "The brightest particles are tetragonal Al3Zr (t-Al3Zr), which in 6086-Sc and 6086-ScY alloys contained these two microalloying elements as well. In addition to these phases, small amounts of minor phases, such as theta-Al2Cu, Q-AlCuMgSi, AlSc2Si2 and ZrSi2, were also identified".  How those phases are identified need to be provided and they should be marked in the figure for direct identification.

We have thoroughly revised the paper, and we hope we were successful in answering these requirements.

  1. A discussion section needs to be provided before the conclusion, and overall presentation needs to be improved.

We provided a discussion section and tried to improve the overall presentation.

  1. The English needs to be checked throughout the manuscript. Multiple grammar errors were caught, for example, "The effect of yttrium on the properties of Al-alloys was much less investigated that 65 the effects of Sc and Zr".

The English were carefully checked. We hope it is acceptable now.

Round 2

Reviewer 1 Report

Authors have incorporated the suggested corrections. The publication of the article is recommended.